# Obatoclax Rescues FUS-ALS Phenotypes in iPSC-Derived Neurons by Inducing Autophagy

**DOI:** 10.3390/cells12182247

**Published:** 2023-09-11

**Authors:** Cristina Marisol Castillo Bautista, Kristin Eismann, Marc Gentzel, Silvia Pelucchi, Jerome Mertens, Hannah E. Walters, Maximina H. Yun, Jared Sterneckert

**Affiliations:** 1Center for Regenerative Therapies TU Dresden (CRTD), Technische Universität Dresden, 01307 Dresden, Germany; cristina_marisol.castillo_bautista@tu-dresden.de (C.M.C.B.); hannah.walters@mailbox.tu-dresden.de (H.E.W.);; 2Core Facility Mass Spectrometry & Proteomics, Center for Molecular and Cellular Bioengineering, Technische Universität Dresden, 01307 Dresden, Germanymarc.gentzel@tu-dresden.de (M.G.); 3Department of Neurosciences, University of California San Diego, La Jolla, CA 92161, USAjerome.mertens@uibk.ac.at (J.M.); 4Department of Pharmacological and Biomolecular Sciences, Università degli Studi di Milano, 20133 Milan, Italy; 5Laboratory of Genetics, The Salk Institute for Biological Studies, La Jolla, CA 92037, USA; 6Institute for Molecular Biology, University of Innsbruck, A-6020 Innsbruck, Austria; 7Max Planck Institute for Molecular Cell Biology and Genetics, 01307 Dresden, Germany; 8Cluster of Excellence Physics of Life, Technische Universität Dresden, 01307 Dresden, Germany; 9Medical Faculty Carl Gustav Carus of TU Dresden, 01307 Dresden, Germany

**Keywords:** phenotypic screening, autophagy, FUS, amyotrophic lateral sclerosis

## Abstract

Aging is associated with the disruption of protein homeostasis and causally contributes to multiple diseases, including amyotrophic lateral sclerosis (ALS). One strategy for restoring protein homeostasis and protecting neurons against age-dependent diseases such as ALS is to de-repress autophagy. BECN1 is a master regulator of autophagy; however, is repressed by BCL2 via a BH3 domain-mediated interaction. We used an induced pluripotent stem cell model of ALS caused by mutant FUS to identify a small molecule BH3 mimetic that disrupts the BECN1-BCL2 interaction. We identified obatoclax as a brain-penetrant drug candidate that rescued neurons at nanomolar concentrations by reducing cytoplasmic FUS levels, restoring protein homeostasis, and reducing degeneration. Proteomics data suggest that obatoclax protects neurons via multiple mechanisms. Thus, obatoclax is a candidate for repurposing as a possible ALS therapeutic and, potentially, for other age-associated disorders linked to defects in protein homeostasis.

## 1. Introduction

Amyotrophic lateral sclerosis (ALS) is an adult-onset disease characterized by progressive loss of upper and lower motor neurons (MNs), leading to progressive paralysis. Death occurs on average within 2–5 years of disease onset [1]. Conventional therapy focuses on symptoms, including respiratory support, and preventing infections. Treatments that effectively protect against ALS neurodegeneration are lacking but urgently needed. Although most ALS cases are sporadic, about 10% show a familial pattern associated with inherited genetic mutations [1], including in the gene Fused-in-Sarcoma (FUS) [2]. FUS encodes a ubiquitously expressed RNA-binding protein involved in DNA damage repair, RNA splicing, RNA transport, translational regulation, and the processing of microRNAs [3]. The FUS protein contains a nuclear localization signal (NLS) domain and is primarily localized to the nucleus. The most prevalent ALS-associated mutations in FUS, including P525L, occur in the NLS domain [4], resulting in FUS mislocalization from the nucleus to the cytoplasm. Importantly, cytoplasmic FUS aggregates are a hallmark of FUS-ALS pathogenesis [5,6]. However, the molecular mechanisms of FUS aggregation and degeneration are not entirely clear. One hypothesis is that stress granules (SGs) might play an important role. Under stress conditions, cytoplasmic FUS is recruited into SGs, altering their composition and causing a very high local concentration of FUS protein [7,8]. Within these aberrant SGs, mutant FUS is thought to undergo a liquid-to-solid phase transition [9,10] that could seed the pathological aggregates observed in patient neurons [11].

Age is one of the most important risk factors for ALS [12], and preventing pathological aging by inducing autophagy might be an effective strategy for delaying the onset of the disease. Autophagy targets cytosolic components such as organelles and protein aggregates to the lysosome [13], and the induction of autophagy by BECN1 is inhibited by forming a repressive complex with multiple anti-apoptotic BH3-domain proteins, including BCL2 (the BECN1-BCL2 complex) [14]. Using a genetic approach to disrupt the BECN1-BCL2 complex, Fernandez and collaborators demonstrated that inducing autophagy counteracts aging, enhances health, and extends lifespan [15]. In addition, this approach rescues the premature aging phenotype displayed by mutant Klotho3 mice, which also manifest neurodegeneration. Consistent with this idea, we previously demonstrated that inducing autophagy reduced cytoplasmic FUS levels and rescued the degeneration of neurons with mutant FUS [8]. Therefore, we hypothesize that drugs inducing autophagy by disrupting the BECN1-BCL2 complex could be effective against ALS.

Here, we aim to identify a small molecule drug that induces autophagy by disrupting the BECN1-BCL2 complex. BH3 mimetics were developed to induce apoptosis in cancer cells by disrupting the BECN1-BCL2 complex [16]. However, the activity of BH3 mimetics on the BECN1-BCL2 complex remains largely unknown. Since the BECN1-BCL2 interaction is mediated by a BH3-domain, we speculate that one of these BH3 mimetics might potently disrupt the BECN1-BCL2 complex, leading to the induction of autophagy, thereby reducing cytoplasmic FUS and protecting neurons against degeneration. Previously, we generated isogenic induced pluripotent stem cell (iPSC) lines in which one FUS allele was tagged with eGFP [8]. iPSC-derived neurons with mutant FUS exhibit altered SG dynamics as well as increased apoptosis [17], recapitulating important aspects of ALS pathogenesis. Using this model, we identified the BH3 mimetic obatoclax as a candidate therapeutic that rescued human iPSC-derived neurons from mutant FUS phenotypes by inducing autophagy via disrupting the BECN1-BCL2 complex.

## 2. Materials and Methods

### 2.1. BH3 Mimetics

The BH3 mimetic compounds ABT-263, ABT-199, ABT-737, AZD5991, Gambogic acid, Gossypol, Obatoclax, Sabutoclax, S55746, and TW-37 (Selleckchem, Cologne, Germany) were dissolved in DMSO to obtain 10 mM stock solutions.

### 2.2. iPSC Culture and Assays

The derivation and characterization of the P525L FUS-eGFP reporter iPSC line were previously described [8]. Cells were differentiated into neurons as previously described [18].
To assay cell survival, neurons were seeded at 4 × 104 cells per well in a 96-well plate. After 20 days, neurons were treated with the compound at different concentrations (1, 10, 100, 1000, and 10,000 nM) for 24 h. After the treatment, the neurons were incubated with calcein-AM red (1 µM, Cayman, Ann Arbor, MI, USA) for 30 min. Then, wash two times with PBS (pH 7.5). The plate was read using a 560 nm excitation filter and a 590 nm emission filter on a Biotek Synergy™ NEO microplate reader (Agilent, Santa Clara, CA, USA). The fluorescence intensity is proportional to the number of viable cells.To assay stress granules, neurons seeded at 4 × 104 cells per well in a 96-well plate were treated with the compound at 10 nM for 24 h. After the treatment, the formation of stress granules was induced with the addition of sodium arsenite (500 mM, Sigma Sigma-Aldrich, St. Louis, MO, USA) for one hour.To monitor autophagic flux, neurons seeded at 4 × 104 cells per well in a 96-well plate and 1.2 × 106 cells per well in a 12-well plate were treated with the compound at 10 nM for 6, 9, 24, and 48 h. Treatment with Bafilomycin A (10 nM, Selleckchem, Cologne, Germany) for 24 h was used as a control.

### 2.3. Protein Extraction, Immunoblotting, and Capillary Electrophoresis

Cell samples were lysed using RIPA buffer supplemented with a protease inhibitor cocktail (Santa Cruz Biotechnology, Dallas, TX, USA). Protein concentration was measured using the Pierce BCA Protein assay kit (ThermoScientific, Waltham, MA, USA). 15–20 µg of protein samples were loaded and separated using SDS-PAGE, followed by wet transfer on a methanol-charged PDVF membrane. The membrane was blocked using 5% milk powder for one hour at room temperature and incubated with the primary antibodies: rabbit Microtubule-associated protein 1A/1B-light chain 3 (LC3) (1:1000, Cat# NB600-1384, Novus Biological, Centennial, CO, USA), mouse BECN1 (1:2000, Cat# ab118148, Abcam, Cambridge, UK), mouse BCL2 (1:1000, Cat# 15071, Cell Signaling, Trask Lane Danvers, MA, USA), rabbit Lysosome-associated membrane glycoprotein 1 (LAMP1) (1:1000, Cat# 9091, Cell Signaling), and rabbit GAPDH (1:4000, Cat# 21185S Cell Signaling). Subsequently, the blots were incubated with either HRP-conjugated anti-rabbit (1:10,000, Cat# 711-035-152, Jackson Immunoresearch, West Grove, PA, USA) or anti-mouse secondary antibodies (1:10,000, Cat# 715-035-150, Jackson Immunoresearch). Blots were developed using ECL detection reagents (GE Healthcare) and visualized with ImageQuant™ LAS 4000 (GE Healthcare, Chicago, IL, USA). Band intensities were quantified using ImageJ.

For capillary electrophoresis, cell lysates were analyzed using the 12–230 kDa separation module associated with the Protein Simple WES™ device. Here, secondary antibodies and reagents were used according to the manufacturer’s instructions (Bio-techne). Cell lysates were loaded at a concentration of 0.4 µg/µL. Primary antibodies were p62 (1:50, Cat# ab56416, Abcam) and GAPDH (1:2000, Cat# 21185S, Cell Signaling).

### 2.4. Proximity Ligation Assay

The proximity ligation assay was performed using the Duolink^®^ In Situ Red Starter Kit Mouse/Rabbit (Sigma-Aldrich, St. Louis, MO, USA) according to the manufacturer’s instructions. 4 × 104 neurons per well on a 96-well plate were fixed for 10 min at room temperature in 4% paraformaldehyde and permeabilized with 0.5% Triton-X 100 (Roth) in PBS for 15 min, followed by three washes with 0.05% Tween-20 (Applichem, Darmstadt, Germany). For blocking and subsequent steps, we followed the provided protocol. Primary antibodies used were BECN1 (1:200, Cat# 66665-1-Ig, Proteintech, Rosemont, IL, USA) and BCL-2 (1:200, Cat# 12789-1-AP, Proteintech). Neurons were imaged using a Zeiss LSM780/FCS laser scanning confocal microscope.

### 2.5. Immunocytochemistry and Image Acquisition

Neurons were fixed for 10 min at room temperature in 4% paraformaldehyde in PBS. Permeabilization and blocking of nonspecific epitopes were performed simultaneously using 0.1% Triton X-100, 1% BSA, and 10% FBS in PBS for 45 min. Subsequently, the primary antibodies mouse anti-tubulin beta 3 (TUBB3) (1:1000, Cat# BLD-801202, Biolegend, San Diego, CA, USA), rabbit cleaved caspase 3 (CC3) (1:400, Cat# 9661S, Cell Signaling), rabbit LC3 (1:1000, Cat# NB600-1384, Novus Biological), and mouse FUS (1:500, Cat# AMAB90549, Sigma) were applied overnight at 4 °C in 0.1% BSA in PBS. The next day, the cells were washed with 0.1% BSA in PBS and incubated with the secondary antibody for 1 h at room temperature. Finally, cells were washed three times with 0.1% BSA in PBS containing 0.005% Tween-20, including Hoechst counter-staining for nuclei in the second washing step. Neurons were imaged with either a Zeiss ApoTome or a Zeiss LSM780/FCS laser scanning confocal microscope as indicated, and image analysis was performed with a Cell Profiler.

### 2.6. Proteomics

iPSC-derived neurons treated with DMSO or obatoclax for 24 h were lysed using RIPA 1x buffer supplemented with a protease inhibitor cocktail (Santa Cruz) and an inhibitor for phosphatase (Roche, Basel, Switzerland). Protein concentration was measured using the Pierce BCA Protein assay kit (Thermo Scientific). A total of 20–25 µg of each sample were separated by SDS gel electrophoresis with a short separation distance of approx. 1.5 cm. Each lane was divided into two slices to limit the gel amount for the in-gel digestion procedure. Proteins were digested in gel according to standard protocols [19,20]. Fractions were combined and analyzed by nanoflow LC-MS/MS with a Dionex3000 RSLC-UPLC hyphenated to a Q Exactive HF mass spectrometer (both Thermo Scientific) operated in DIA mode. The MS data were processed with DIA-NN V1.8 [21]. Methodical details are provided in the supplement (Appendix A). The proteomics data were deposited at the PRIDE database [22], a member of the ProteomeXchange consortium [23] for proteomic data (EBI, Cambridge, UK), under an accession number that will be provided before acceptance of the manuscript). Differential expression analysis of the proteins was performed using the DEP package in R [24]. The threshold for the differential expression was ±1.5 fold change with an adjusted *p*-value of 0.05. KEGG enrichment analysis was performed using EnrichR [25,26].

### 2.7. Statistical Analysis

Prism 8.0 software (GraphPad) was used for all statistical analysis. Comparison between two groups (untreated vs. treated) was carried out by an unpaired two-tailed t test. A *p*-value lower than 0.05 was considered significant.

## 3. Results

### 3.1. Obatoclax Is Well-Tolerated and Potently Reduces Aberrant SG Formation

Disrupting the BECN1-BCL2 complex induces autophagy, thereby increasing health and lifespan. Previously, we demonstrated that inducing autophagy protected human iPSC-derived neurons against mutant FUS, leading us to speculate that disrupting the BECN1-BCL2 interaction in human neurons could be an effective strategy to protect them against FUS-ALS. BH3 mimetics disrupt BCL2-interactions. Therefore, we sought to identify a BH3 mimetic that protects human iPSC-derived neurons against FUS-ALS pathogenesis by inducing autophagy.

Previously, we modeled FUS-ALS by generating P525L FUS-eGFP iPSCs. Neurons differentiated from this iPSC line spontaneously fire tetrodotoxin-sensitive action potentials (Appendix A) and recapitulate aspects of ALS pathology [8]. Since our objective was to protect neurons against ALS-associated degeneration, it was important that candidate compounds not be toxic to iPSC-derived neurons. BCL2 is an important regulator of apoptosis, and BH3 mimetics are known to be cytotoxic and are even in clinical use to target cancer cells [16]. Thus, our first experiment was to characterize the cytotoxicity of ten selected BH3 mimetics using iPSC-derived neurons with P525L FUS-eGFP: ABT-263, ABT-199, ABT-737, TW-37, obatoclax, AT101, sabutoclax, AZD5991, S55746, and gambogic acid. Cell viability was quantified using calcein-AM Red after treatment with compounds for 24 h at concentrations of 1 nM to 10 µM (Figure 1a and Appendix A). We found that most compounds are well tolerated at 10 nM. ABT-263 and obatoclax showed significant toxicity at 100 nM. ABT-737 and gambogic acid showed significant toxicity at 1 µM. ABT-199, TW-37, and S55745 showed significant toxicity at 10 μM. AZD5991 was well tolerated at all tested concentrations. The maximum tolerated dose of each compound was selected for further analysis.

Aberrant SGs are thought to play an important role in ALS, including FUS-ALS [27]. When exposed to a stressor such as arsenite, iPSC neurons with mutant FUS-eGFP form SGs containing high levels of FUS-eGFP, altering SG size and number compared to isogenic controls [8]. We characterized the effects of treating iPSC-derived neurons for 24 h with the maximum concentration well tolerated for each compound on P525L FUS-eGFP-positive SG formation (500 μM arsenite for 1 h). ABT-737 and gambogic acid significantly decreased the number of P525L FUS-eGFP-positive SGs at 100 nM (Figure 1b). However, obatoclax was particularly interesting because we observed a significantly reduced number of P525L FUS-eGFP-positive SGs with only 10 nM (Figure 1b), and additional testing revealed a similar effect with only 1 nM (Appendix A). We observed no toxicity, including to senescent cells, at 10 nM (Appendix A). Interestingly, we observe that obatoclax treatment decreases senescent cell viability at high concentrations (about 100 nM) in a manner comparable to ABT-263, highlighting its potential as a senolytic compound. It is also important to note that obatoclax crosses the blood-brain barrier [28] and has been tested in phase II clinical trials as an anti-cancer drug, although at higher concentrations in order to induce apoptosis in cancer cells [29,30,31]. For these reasons, obatoclax was selected for further analysis.

### 3.2. Obatoclax Reduces Cytoplasmic FUS Levels, Rescuing Aberrant Protein Homeostasis

Mutations in the FUS NLS, such as P525L, cause ALS and increase the cytoplasmic levels of FUS, leading to defects in protein homeostasis as well as neurodegeneration [32]. In addition, we demonstrated that increasing autophagy via mTOR inhibition promoted the clearance of cytoplasmic FUS [17]. Thus, we characterized the impact of obatoclax on the cytoplasmic levels of P525L FUS-eGFP in iPSC-derived neurons. Unfortunately, obatoclax is an autofluorescent compound, and its emission overlaps with FUS-eGFP [33]. For this reason, we used immunostaining to quantify cytoplasmic FUS levels. We found that treatment of P525L FUS iPSC-derived neurons with obatoclax at 10 nM for 24 h significantly decreased the level of cytoplasmic FUS compared with DMSO (Figure 2a). Nuclear FUS was not significantly altered.

It has been shown that iPSC-derived neurons with mutant FUS manifest an accumulation of p62, which indicates that cytoplasmic FUS is associated with defects in protein homeostasis [17,34]. Since obatoclax reduced cytoplasmic FUS levels, we next tested if obatoclax would rescue protein homeostasis as marked by p62 levels. We assessed p62 protein levels via WES capillary electrophoresis and found that iPSC-derived neurons with P525L FUS-eGFP treated with 10 nM obatoclax for 24 h showed decreased p62 levels compared with DMSO (Figure 3), which indicates that obatoclax ameliorates defects in protein homeostasis in iPSC-derived neurons with mutant FUS.

### 3.3. Obatoclax Ameliorates the Degeneration of iPSC-Derived Neurons with Mutant FUS

FUS-ALS patients suffer paralysis due to the loss of MNs, and previously our group reported that P525L FUS-eGFP iPSC-derived neurons manifest increased degeneration as marked by the apoptosis marker cleaved-caspase 3 (CC3) [8,17]. Although high concentrations of obatoclax induce apoptosis, we unexpectedly found that at concentrations as low as 1 nM, obatoclax unexpectedly ameliorates mutant FUS-associated phenotypes. Nevertheless, the ability of obatoclax to induce apoptosis makes it particularly important to characterize the effects of low nM concentrations of obatoclax on neurodegeneration as marked by CC3. We observed that P525L FUS-eGFP iPSC-derived neurons treated with 10 nM obatoclax for 24 h showed lower levels of CC3 compared with controls (Figure 2b). It is important to note that obatoclax’s neuroprotective effects are observed at concentrations (1–10 nM) that are at least two orders of magnitude less than the concentrations (0.2–1 μM) used to induce cell death in cancer cell lines [29,30,31]. Thus, our results suggest that there is a therapeutic window for using obatoclax as a possible ALS therapeutic. However, more work is needed, including in mouse models, before initiating clinical trials.

### 3.4. Obatoclax Induces Autophagy by Disrupting the BECN1-BCL2 Complex

We speculated that obatoclax rescued mutant FUS-associated phenotypes in iPSC-derived neurons by inducing autophagy. Thus, we evaluated the ability of obatoclax to induce autophagy in mutant FUS iPSC-derived neurons. LC3-II levels were quantified using Western blot on iPSC-derived neurons treated with obatoclax for increasing amounts of time (6, 9, 24, and 48 h). We found that the levels of LC3B-II protein increased gradually (maximum level in 24 h) and then decreased at 48 h (Figure 4a), demonstrating that obatoclax alters autophagy.

To distinguish whether the decrease of LC3-II at the 48-h time point is due to a reduction in the production of autophagosomes or due to increased autophagic flux, we examined obatoclax-stimulated autophagosome formation in the presence and absence of Bafilomycin A1 (BafA1), which inhibits fusion of autophagosomes with lysosomes as well as lysosome acidification [35]. Obatoclax was relatively stable in solution for 48 h, and resupplying obatoclax every 24 h yielded similar results as one dosage for 48 h (Appendix A). BafA1 in combination with obatoclax increased LC3-II protein levels compared with those treated with obatoclax alone (Figure 4b,c), suggesting that autophagic flux is indeed stimulated by obatoclax. In addition, we measured the protein levels of LAMP1, which marks lysosomes, and we found a trend toward increased LAMP1 protein levels at 48 h (Figure 4d). Finally, we performed LC3 immunostaining to visualize the autophagosomes and observed a significant increase in LC3 puncta in iPSC-derived neurons treated with obatoclax at 24 h (Figure 5). Taken together, these results demonstrate that obatoclax induces autophagy in P525L FUS-eGFP iPSC-derived neurons.

Since obatoclax is a BH3 mimetic, we hypothesized that obatoclax induces autophagy by disrupting the interaction between BCL2 and BECN1. To test this hypothesis, we performed the proximity ligation assay, which revealed a decrease in the interaction of BECN1 and BCL2 in P525L FUS-eGFP iPSC-derived neurons treated with obatoclax at 10 nM for 24 h compared with DMSO (Figure 6). This result demonstrates that obatoclax induces autophagy by disrupting the BCL2 interaction with BECN1, consistent with being a BH3 mimetic. Therefore, we conclude that obatoclax ameliorates ALS phenotypes in mutant FUS iPSC-derived neurons, including reducing cytoplasmic FUS levels, reducing aberrant SG formation, ameliorating defects in protein homeostasis, and reducing degeneration, by inducing autophagy via disruption of BCL2-mediated inhibition of BECN1.

### 3.5. Proteomics Suggests Obatoclax Contributes to Neuroprotection via Multiple Mechanisms

To better understand the mechanism by which obatoclax ameliorates mutant FUS-associated phenotypes in iPSC-derived neurons, we performed label-free proteomics. Specifically, we compared WT FUS-eGFP neurons with P525L FUS-eGFP neurons as well as P525L FUS-eGFP neurons treated with either DMSO or obatoclax. 6696 proteins were identified in the samples, of which 6531 were quantifiable (Appendix A). The principle component analysis (Figure 7a) showed two different clusters that correspond to WT FUS-eGFP and P525L FUS-eGFP neurons. In comparison with WT FUS-eGFP neurons, 56 proteins were increased and 56 decreased in P525L FUS-eGFP neurons (Figure 7b). The Kyoto Encyclopedia of Genes and Genomes (KEGG) (Appendix A) pathway enrichment analysis identified only one term as being significantly enriched: proteasome (*p* = 0.000126). Three proteasome-associated proteins, PSME1, PSME2, and PSME3, are downregulated in P525L FUS-eGFP neurons compared with WT (Figure 7d), suggesting that mutant FUS significantly alters proteasome activity in iPSC-derived neurons. This is consistent with previous reports [36,37,38,39,40], which have shown an association between decreased proteasome activity and ALS pathogenesis. Thus, increasing autophagy using obatoclax might help alleviate some of the stress caused by aberrant proteasome regulation.

Next, we analyzed the effects of obatoclax. A total of 17 proteins were increased and 4 decreased by obatoclax in P525L FUS-eGFP neurons and iPSC-derived neurons compared to DMSO (Figure 7c). KEGG pathway enrichment showed that only one term was enriched: Mucin type O-glycan biosynthesis (*p* = 0.000647). One of the two proteins associated was ST3GAL1, which is critical for the biosynthesis of the ganglioside GM1 [41] and is downregulated in mutant FUS compared with WT FUS. ST3GAL1 protein levels are rescued in mutant FUS neurons treated with obatoclax (Figure 7e). B4GALT5 is another protein involved in the biosynthesis of the gangliosides [42] and is upregulated in mutant FUS treated with obatoclax (Figure 7e). This is important because GM1 is an important factor in maintaining the mammalian central nervous system and preventing neurodegeneration [43,44]. One report showed increased anti-GM1 autoantibodies in ALS patients, which could suggest that increasing GM1 could be protective against age-associated degenerative disorders such as ALS [45]. Consistent with this idea, supplementation with GM1 improved the spatial learning and memory of aged rats [46].

It is also interesting to note that obatoclax ameliorated the effects of P525L FUS-eGFP on several proteins that are important regulators of neuronal survival (Figure 7f,h). EHD4 is implicated in endocytic trafficking, especially in early endosomes [47], and AP1S1 is involved in protein sorting in the Golgi network and endosomes [48]. MRGBP inhibits DNA double-strand break repair [49], and DNAJC9 is both a histone chaperone as well as a heat shock-induced chaperone [50]. Each of these processes has been causally linked to neurodegeneration in ALS [51,52,53,54,55,56,57,58]. Our data suggest that obatoclax could contribute to the protection of neurons against ALS pathogenesis via multiple mechanisms.

## 4. Discussion

Aging is a significant risk factor for ALS and other neurodegenerative diseases. Genetically disrupting the BECN1-BCL2 complex induces autophagy in vivo independent of the mTOR pathway, leading to increased health and lifespan [15,59,60]. This suggests that disrupting the BECN1-BCL2 complex could be an effective therapeutic strategy for ALS. Since the BECN1-BCL2 interaction is mediated by a BH3 domain, we screened BH3 mimetics for their ability to ameliorate mutant FUS-associated phenotypes in iPSC-derived neurons. We identified obatoclax as a compound that reduced mutant FUS-eGFP stress granules at concentrations as low as 1 nM and had the capability to cross the blood-brain barrier [28].

Obatoclax was developed as a pan-BH3 mimetic to induce apoptosis of cancer cells by disrupting multiple interactions, including those between anti-apoptotic BCL2 family proteins (BCL2, BCLXL, and MCL1) and BAX and BAK interactions [61]. To inhibit so many complexes, obatoclax has previously been characterized at high concentrations (0.2–1 μM), at which it is cytotoxic [29,30,31]. However, disrupting a single interaction, namely BECN1-BCL2, induces autophagy, a considerably more specific target than disrupting all BH3 interactions to induce apoptosis. This difference could explain why obatoclax induced autophagy, reduced cytoplasmic FUS, and decreased neuronal cell death at only 10 nM, which is one to two orders of magnitude less than the concentration at which cytotoxicity is observed. This could have important clinical implications as well.

Obatoclax rescues mutant FUS-associated phenotypes in iPSC-derived neurons. Our proteomics analysis suggested that the proteasome may not be functioning normally in neurons with P525L FUS-eGFP, which is consistent with a previous report showing impairment of proteasome function in sporadic ALS [39]. In addition, failure of protein quality control, to which the proteasome is a critical contributor, is a hallmark of ALS [62]. Therefore, we speculate that cytoplasmic FUS might impair the activity of the proteasome in P525L FUS-eGFP neurons, which could explain why increasing autophagy using obatoclax would have a beneficial effect.

Our proteomics analysis suggests that obatoclax could contribute to neuroprotection via multiple mechanisms, including increased biosynthesis of gangliosides such as GM1. Importantly, GM1 ganglioside is linked to both aging and autophagy. GM1 ganglioside induces autophagy and neuroprotection in models of Parkinson’s disease [63] and Alzheimer’s disease [64]. We found that neurons with mutant FUS showed decreased ST3GAL1, an enzyme involved in GM1 ganglioside biosynthesis. Obatoclax restored ST3GAL1 protein levels to WT levels. It is also interesting to note that obatoclax significantly altered the levels of proteins involved in endosomal trafficking, the Golgi network, DNA damage, and chaperone activity, each of which have been causally linked to ALS pathogenesis [51,52,54,55,56,65], suggesting that obatoclax protects neurons via multiple mechanisms.

Obatoclax has been evaluated preclinically and clinically in both hematological malignancies and solid tumors. It has been reported that in animal models, obatoclax did not provoke weight loss or other indicators of generalized toxicity [66,67,68]. However, obatoclax leads to neurological toxicity at high doses [67,68]. In clinical trials in cancer patients, the drug was well tolerated, producing predominantly transient, mild central nervous system side effects. Those side effects were more frequent at higher doses in shorter infusions [28,69,70]. However, those trials were aimed at inducing apoptosis in cancer cells. We observed the rescue of mutant FUS phenotypes at a concentration as low as 1 nM, and, at this lower dose, the only adverse effect reported was diarrhea [70]. Thus, our work identifies obatoclax as a potential drug candidate for repurposing to target ALS by inducing autophagy. Since BECN1-BCL2 regulates age-associated decreases in health span, it could be interesting to characterize the effects of obatoclax on other age-associated diseases.

## 5. Conclusions

This work provides evidence that obatoclax ameliorates mutant FUS-associated ALS phenotypes in human neurons by inducing autophagy by disrupting the BECN1-BCL2 complex. Our data demonstrate that obatoclax reduces cytoplasmic FUS, restores protein homeostasis, and reduces degeneration of neurons with mutant FUS. Thus, obatoclax is a candidate for drug repurposing for ALS; however, additional experiments, including in vivo testing, are required to evaluate its therapeutic potential.

## Figures and Tables

**Figure 1 cells-12-02247-f001:**
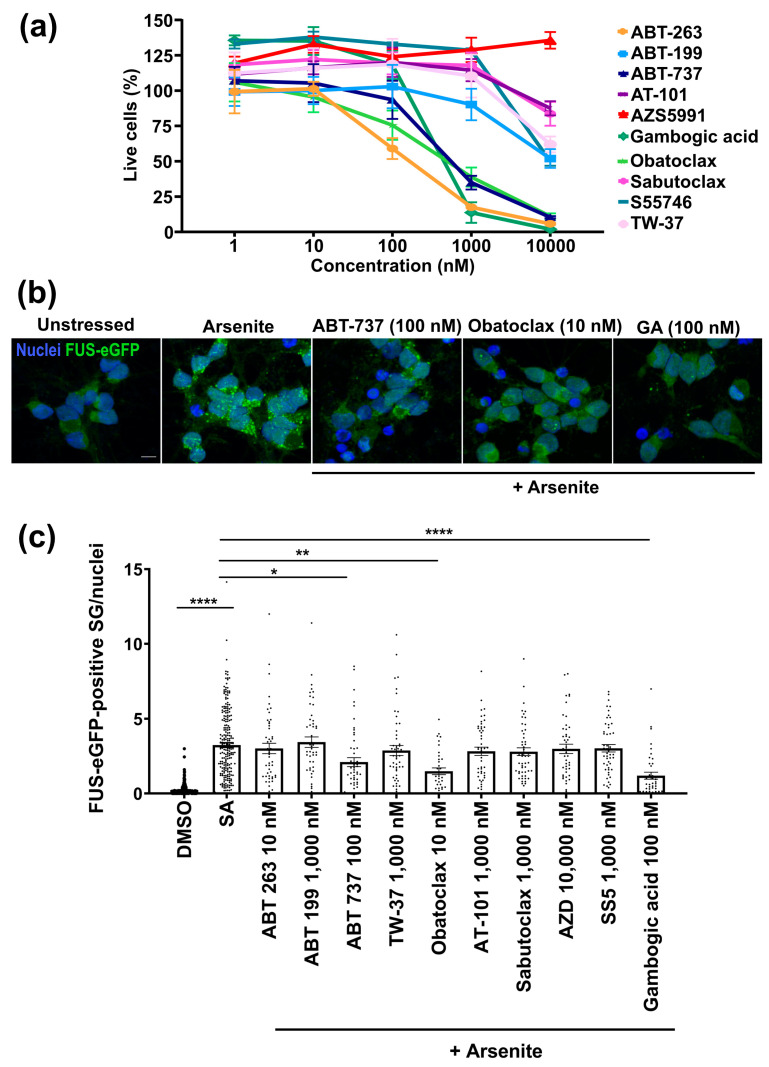
Identification of BH3 mimetic compounds reducing the P525L FUS-eGFP SG phenotype in iPSC-derived neurons. (**a**) Effect on the cell viability of the compounds in iPSC-derived neurons. Compounds were tested at different concentrations for 24 h. Cell viability was assessed with calcein AM-Red. The mean of three independent experiments (n = 3) is shown; error bars indicate the standard error of the mean (SEM). Related to Appendix A. (**b**) Compounds were tested for 24 h at concentrations that were well tolerated. Fluorescent confocal micrographs show that the indicated compounds reduce FUS-eGFP-positive SGs in iPSC-derived neurons treated for 24 h. GA = gambogic acid. Scale bar = 10 μm. (**c**) Quantification of individual FUS-eGFP-positive SGs from three independent experiments (n = 3); error bars indicate SEM. Significance was tested using the Kruskal-Wallis test with a Dunn post-test. **** denotes the significance of *p* < 0.0001 between unstressed and arsenite. **** denotes the significance between treatments and arsenite. *, ** and **** indicate *p* < 0.5, <0.01, and <0.0001, respectively. Related to Appendix A.

**Figure 2 cells-12-02247-f002:**
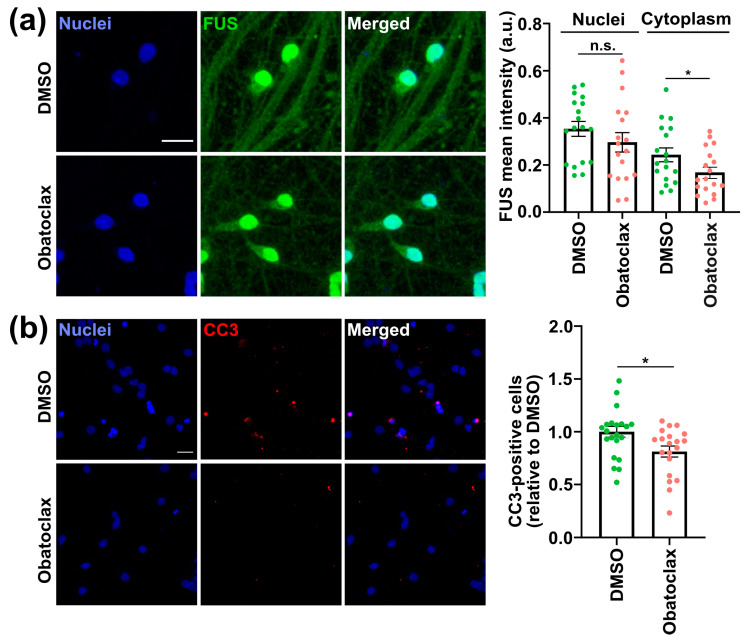
Obatoclax is associated with ALS-associated phenotypes in P525L FUS-eGFP iPSC-derived neurons. (**a**) Levels of cytoplasmic and nuclear FUS-eGFP in iPSC-derived neurons treated with DMSO and obatoclax at 10 nM for 24 h. The mean of six independent experiments (n = 6) is shown; error bars indicate SEM. Treatments were analyzed via an unpaired student’s *t*-test. * indicates *p* < 0.05. n.s. indicates not significant. Scale bar = 20 μm. (**b**) Cleaved caspase 3 (CC3) levels in iPSC-derived neurons treated with DMSO and obatoclax at 10 nM for 24 h. The mean of six independent experiments (n = 6) is shown; error bars indicate SEM. Treatments were analyzed via an unpaired student’s t-test. * indicates *p* < 0.05. Scale bar = 20 μm. Related to Appendix A.

**Figure 3 cells-12-02247-f003:**
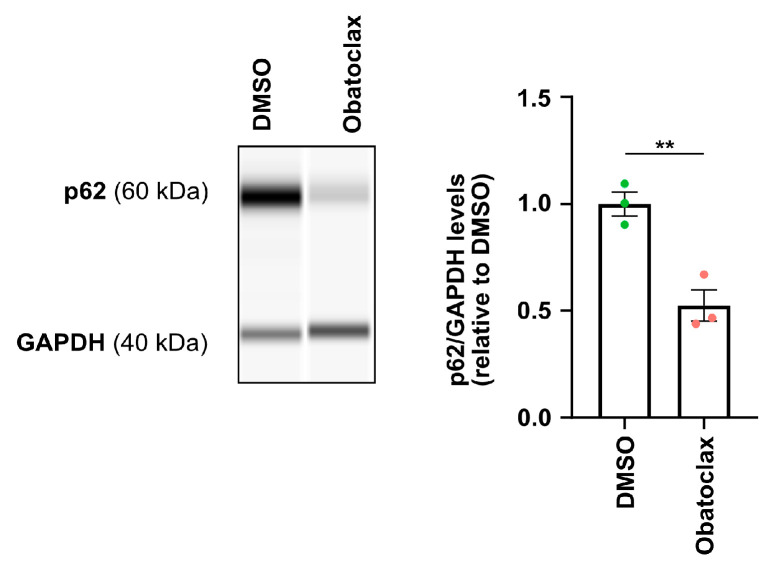
Obatoclax restores protein homeostasis in P525L FUS-eGFP iPSC-derived neurons. p62 protein levels in iPSC-derived neurons treated with DMSO and obatoclax at 10 nM for 24 h. The mean of three independent experiments (n = 3) is shown; error bars indicate SEM. Treatments were analyzed via an unpaired student’s *t*-test. ** indicates *p* < 0.01.

**Figure 4 cells-12-02247-f004:**
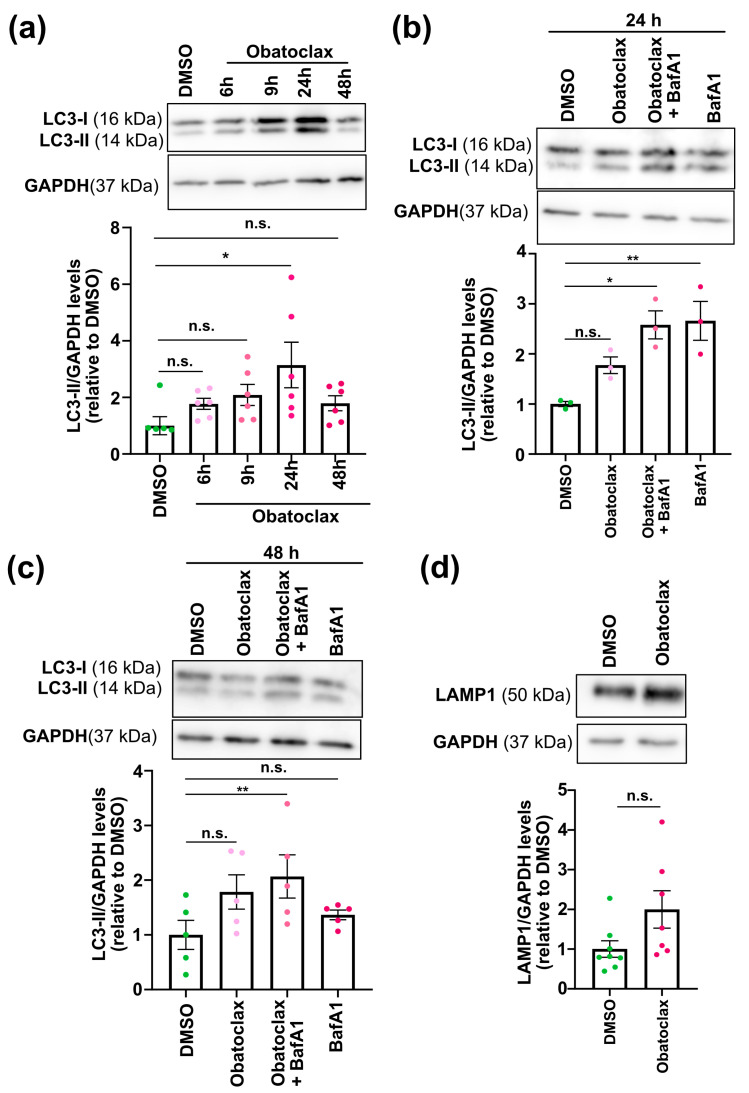
Obatoclax induces autophagy in P525L FUS e-GFP iPSC-derived neurons. (**a**) Obatoclax was tested at 10 nM for 6, 9, 24, and 48 hours. We assessed the LC3B-II protein levels using Western blotting. Results show the mean of six independent experiments (n = 6); error bars indicate SEM. Treatments were analyzed via one-way ANOVA with a Dunnett post-test, * indicates *p* < 0.5; n.s. indicates not significant. (**b**) Obatoclax was tested at 10 nM for 24 hours with and without Bafilomycin A at 10 nM for 24h. We assessed the LC3B-II protein levels using Western blotting. Results show the mean of three independent experiments (n = 3); error bars indicate SEM. Treatments were analyzed via one-way ANOVA with a Dunnet post-test; * indicates *p* < 0.05. ** indicates *p* < 0.01. n.s. indicates not significant. BafA1 = Bafilomycin A1. (**c**) Obatoclax was tested at 10 nM for 24 hours with and without Bafilomycin A at 10 nM for 48 hours. We assessed the LC3B-II protein levels using Western blotting. Results show the mean of five independent experiments (n = 5); error bars indicate SEM. Treatments were analyzed via one-way ANOVA with a Dunnet post-test, ** indicates *p* < 0.01. n.s., not significant. BafA1 = Bafilomycin A1. (**d**) Obatoclax was tested at 10 nM for 24 hours. We assessed the LAMP1 protein using Western blotting. Results show the mean of eight independent experiments (n = 8); error bars indicate SEM. n.s. indicates not significant. Related to Appendix A.

**Figure 5 cells-12-02247-f005:**
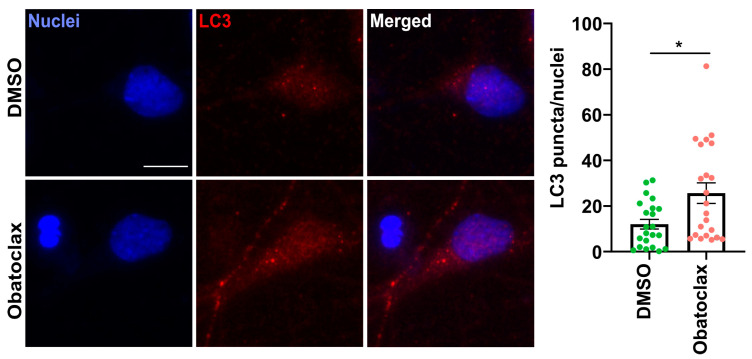
Obatoclax induces autophagy in P525L FUS-eGFP iPSC-derived neurons. Confocal fluorescent micrographs showing LC3 puncta in P525L FUS-eGFP iPSC-derived neurons treated with DMSO and obatoclax at 10 nM for 24 hours. Obatoclax was tested at 10 nM for 24 hours. We assessed the LC3B-II protein levels using Western blotting. The mean of three independent experiments (n = 3); error bars indicate SEM. Treatments were analyzed via an unpaired student’s *t*-test. * indicates *p* < 0.05. Scale bar = 10 μm.

**Figure 6 cells-12-02247-f006:**
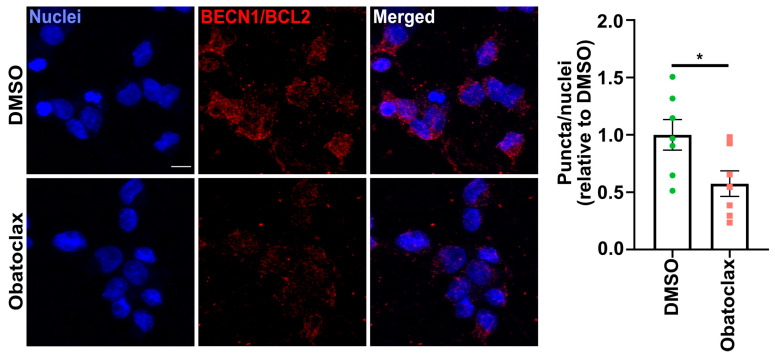
Obatoclax disrupts the BECN1-BCL2 complex in iPSC-derived neurons. Proximity ligation assay to assess the interaction of BECN1 and BCL2 in iPSC-derived neurons treated with DMSO and obatoclax at 10 nM for 24 h. The mean of eight independent experiments (n = 8); error bars indicate SEM. Treatments were analyzed via an unpaired Student’s *t*-test. * indicates *p* < 0.05. Scale bar = 10 μm.

**Figure 7 cells-12-02247-f007:**
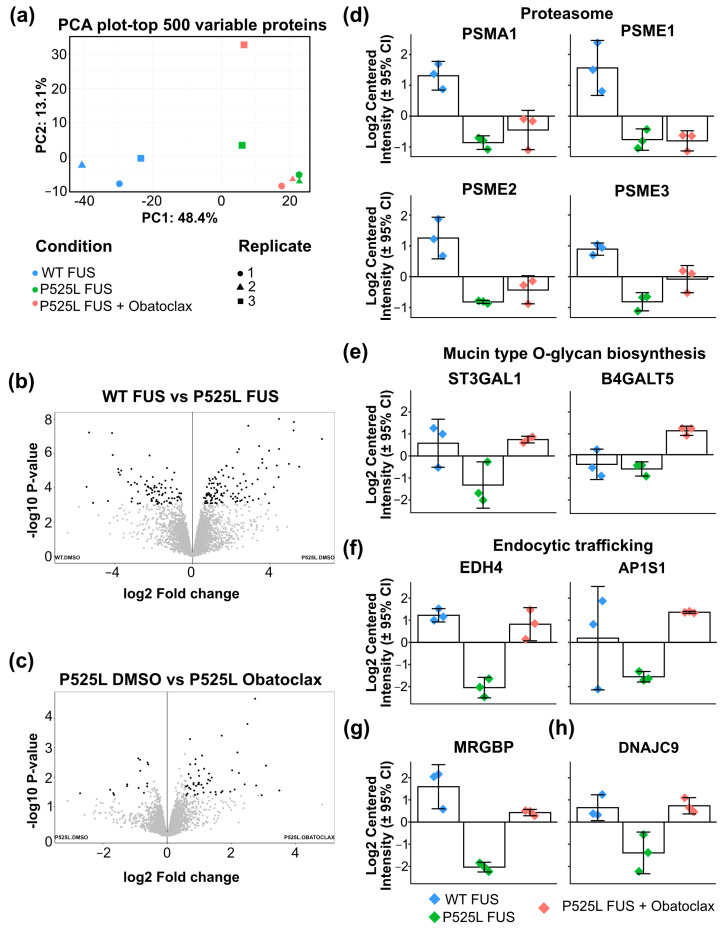
Proteomic analysis of iPSC-derived neurons. (**a**) Principle component analysis (PCA) plot of the proteome of the samples analyzed: WT FUS-eGFP treated with DMSO, P525L FUS-eGFP treated with DMSO, and P525L FUS-eGFP treated with obatoclax at 10 nM for 24 h. Three independent experiments. See also Appendix A. (**b**) Volcano plot of the comparison between WT FUS-eGFP treated with DMSO and P525L FUS-eGFP treated with DMSO. (**c**) Volcano plot of the comparison between P525L FUS treated with DMSO and P525L FUS-eGFP treated with obatoclax at 10 nM for 24 h. (**d**) Differentially expressed proteins related to the proteasome pathway. Proteasome subunit alpha type-1 (PSMA1, *p* = 0.000103), proteasome activator complex subunit 1 (PSME1, *p* = 0.000251), proteasome activator complex subunit 2 (PSME2, *p* = 0.000132), and proteasome activator complex subunit 3 (PSME3, *p* = 0.000054) are downregulated in P525L FUS-eGFP compared with WT FUS-eGFP. (**e**) Differentially expressed proteins related to the mucin type O-glycan biosynthesis pathway. Lactosylceramide alpha-2,3-sialyltransferase (ST3GAL1, *p* = 0.005826) and beta-1,4-galactosyltransferase 5 (B4GALT5, *p* = 0.000386) are upregulated in P525L FUS-eGFP treated with obatoclax compared with P525L FUS-eGFP treated with DMSO. (**f**) Proteins related to endocytic trafficking. EH domain-containing protein 4 (EHD4, 0.000021) and AP-1 complex subunit sigma-2 (AP1S1, *p* = 0.004056) are upregulated in P525L FUS-eGFP treated with obatoclax compared with P525L FUS-eGFP treated with DMSO. (**g**) MRG/MORF4L-binding protein (MRGBP, *p* = 0.000154) related to DNA double-strand break repair. (**h**) DnaJ homolog subfamily C member 9 (DNAJC9, *p* = 0.001442) related to histone chaperone network and heat shock-induced response. Related to Appendix A.

## Data Availability

The data for this study are available from the corresponding authors upon reasonable request.

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
