# Peer review of "Obatoclax Rescues FUS-ALS Phenotypes in iPSC-Derived Neurons by Inducing Autophagy"

_cells, 2023, doi:10.3390/cells12182247_

Round 1

Reviewer 1 Report

The authors present a study on iPSC-derived neurons  where obatoclax rescued FUS-phenotype. The manuscript is clear and presented in a well-structured manner, the experimental design is appropriate and the study can help in treatment research for ALS. The only one issue is the discussion that should be widened for instance discussing how these results could be importnt also for non-FUS ALS patients. 

Author Response

We thank the reviewer for their support. As requested, we have modified the discussion to include how these results could be important for other ALS subtypes (see lines 458-459).

Reviewer 2 Report

In this paper, the authors studies on identifying potential treatment drug for ALS. The authors used an iPSC derived neuron culture model carrying ALS related mutation on FUS gene to screen a set of BH3 mimetics, which based on the molecular mechanism of autophagy in ALS, should elevate the deficits on protein homeostasis and apoptosis. They identified obatoclax, which is a brain-blood-barrier-penetrant small molecule in clinical trial, that can de-repress autophagy and rescue the phenotypes. In general, the authors demonstrated the effect of obatoclax autophagy and apoptosis in neurons and provided potential strategy for ALS treatment.

I have a few minor concerns:

1. Figure 1a. for many treatments, the live cell percentage are higher than 100%.The author could provide more information on how the numbers were obtained, what is the normalization method. The authors could also add the DMSO control group data and statistics.

2. Based on the method, the neurons were treated at day 20. At this stage, how mature are these neurons comparing to the neurons in patients at the onset of the disease?

3. Figure 2b. The authors found that at low concentration, the obatoclax has neuroprotective effect. Did the authors check whether high concentration of obatoclax could recapitulate the neurotoxicity in their differentiated neuron model?

4. Figure 4a. ‘the levels of LC3B-II protein in- 286 creased gradually (maximum level in 24 hours) and then decreased at 48 hours.’ Did the author know the stability of obatoclax in cell culture? Could the decrease at 48 hours be degradation/depletion of obatoclax? If so, should the authors resupply the drug after 24 hours?

Author Response

We thank the reviewer for their time and support.

Point 1: We agree that this is a bit confusing. The normalized numbers were obtained by comparing each sample against DMSO alone with the respective concentrations. One possible reason for variation in the number of cells seeded per well at time 0, or, possible, some kind of unexpected interaction between the specific compound and the assay readout. To help clarify things, we have included the DMSO data points. In addition, we have included Supplementary Figure S4. Cell viability of the BH3 mimetics compounds. Related to Figure 1, which shows the DMSO data points and the statistical analysis.

Point 2: We have performed multi-electrode array analysis of iPSC-derived neurons, which demonstrated that neurons spontaneously fired tetrodotoxin-sensitive action potentials (Figures S1-S3).

Point 3: As the reviewer points out, these compounds are known to be toxic. That is why our first experiment was to identify a concentration that was well tolerated by iPSC-derived neurons. In figure 1, we show that 100 nM (or higher) obatoclax is neurotoxic.

Point 4: Identifying the stability of obatoclax in cell culture is difficult. The compound is internalized by cells, which means that it would be necessary to purify obatoclax from whole cell lysates. Unfortunately, this is quite difficult and beyond the scope of this study. Instead, we quantified the stability of 100 µM obatoclax in solution in PBS at 37 °C for 24 and 48 hours. After 24 hours, we observed that over 70% of obatoclax is still in solution and after 48h we observed that over 50% is still in solution (added as supplemental figure S7), suggesting that obatoclax is relatively stable. In addition, it is important to note that obatoclax is still effective even at a concentration as low as 1 nM (Figure S5). Thus, even if 90% of obatoclax is degraded, we would still expect to have an effect on the cells. For this reason, we believe that the LC3B-II decrease is due to clearance of autophagosomes via lysosomes (e.g., autophagic flux) (Figure 4D). We have also added an experiment showing that neurons treated with obatoclax for 48 hours harbored the same number of LC3 puncta as neurons treated twice (at 0 and 24 hours) (Figure S8).